# Electron addition spectral functions of low-density polaron liquids

Alberto Nocera[*1,2]
Mona Berciu[**1,2,3]

[1] Stewart Blusson Quantum Matter Institute, University of British Columbia,
Vancouver, British Columbia, V6T 1Z4 Canada
[2] Department of Physics Astronomy, University of British Columbia,
Vancouver, British Columbia, Canada V6T 1Z1
[3] Leibniz Institute for Solid State and Materials Research (IFW) Dresden,
Helmholtzstrasse 20, 01069 Dresden, Germany
* alberto.nocera@ubc.ca ** berciu@phas.ubc.ca

April 4, 2023

## Abstract

Spectral functions are important quantities that contain a wealth of information about the quasiparticles of a system, and that can also be measured experimentally. For systems with electron-phonon coupling, good approximations for the spectral function are available only in the Migdal limit (at Fermi energies much larger than the typical phonon frequency, $E_F \gg \Omega$, requiring a large carrier concentration $x$) and in the single polaron limit (at $x = 0$). Here we show that the region with $x \ll 1$ ($E_F < \Omega$) can also be reliably investigated with the Momentum Average (MA) variational approximation, which essentially describes the formation of a polaron above an inert Fermi sea. Specifically, we show that for the one-dimensional spinless Holstein model, the MA spectral functions compare favorably with those calculated using variationally exact density matrix renormalization group simulations (DMRG) evaluated directly in frequency-space, so long as $x < 0.1$ and the adiabaticity ratio $\Omega/t > 0.5$. Unlike in the Migdal limit, here 'polaronic physics' emerges already at moderate couplings. The relevance of these results for a spinful low-$x$ metal is also discussed.

# 1   Introduction

A charged carrier polarizes its surroundings, dressing itself with a cloud of bosonic excitations of the host crystal including phonons and charge, spin and/or orbital electronic fluctuations. The properties of the resulting quasiparticle (when one is well defined) are of central practical interest because they play a key role in controlling the macroscopic behaviour of materials. Solving such problems is also of tremendous fundamental interest because they provide a challenge for any formalism aiming to span weak, intermediate and strong coupling regimes—their study has already led to many important developments in many-body physics and quantum field theory.

An important subset of such problems is the study of polarons, which are the quasiparticles of systems with electron-phonon coupling. To calculate their spectral weights, which can be directly measured with angle resolved photoemission spectroscopy, requires finding the electron-addition propagator [1]

$$G(k, z) = \langle GS_{N_e}|c_{k\sigma}\hat{G}(z + E_{GS,N_e})c_{k\sigma}^{\dagger}|GS_{N_e}\rangle \tag{1}$$

where $\hat{G}(z) = [z - \mathcal{H}]^{-1}$ is the resolvent of the Hamiltonian $\mathcal{H}$ of interest, $N_e = xN$ is the number of electrons set by the electron concentration $x$ and the number of unit cells $N \to \infty$, $\mathcal{H}|GS_{N_e}\rangle = E_{GS,N_e}|GS_{N_e}\rangle$ is the ground-state with $N_e$ electrons, and $c_{k\sigma}^{\dagger}$ creates an electron with momentum $k$ and spin $\sigma$. We note that an electron-removal propagator can be defined similarly. [1] For the example considered below, it is obtained from the electron-addition propagator by replacing $\omega \to -\omega, x \to 1 - x$.

Despite considerable efforts and some significant successes in the $\sim 90$ years since Landau opened this field, [2] good general knowledge about the spectral functions of such systems is available only in two small regions of the parameter space: (i) single polarons, and (ii) the Migdal limit.

The study of single polarons, at $N_e = 0$, is simpler because $|GS_{N_e=0}\rangle \equiv |0\rangle$ is the phonon vacuum and $E_{GS,N_e=0} = 0$. Moreover, the absence of other carriers render polaron-polaron interactions irrelevant. Single polarons have been studied for a variety of electron-phonon couplings, the most famous being the Holstein, [3,4] Fröhlich [5] and Peierls/SSH [6–9] models. Many theoretical methods were developed to calculate the propagator of Eq. (1), including perturbation theory, [10] semiclassical approximations, [2,11] dynamical mean-field theory, [12,13] path integral techniques, [14,15] and variational methods [16] including the momentum average (MA) approximation. [17–20] These were supplemented by a wide variety of computational techniques [21] such as variational exact diagonalization [22,23] and various types of quantum Monte Carlo simulations, [24–28] plus the density matrix renormalization group (DMRG) in one dimension. [29,30] By and large, there is now a good understanding of the generic properties of single polarons of model Hamiltonians at all couplings from very weak to very strong, and many reliable tools to investigate new models. [31] More recently,

significant efforts have been devoted to combining these various tools with first principles calculations for an accurate description of single polarons in actual materials. [32–34]

Generalizing these methods to study the spectral function of polarons in systems with finite carrier concentration $x$ is difficult, first because the ground state $|GS\rangle_{N_e}$ now describes a complicated liquid of polarons or bipolarons (*i.e.* two carriers bound through exchange of phonons) [23,35] and might exhibit superconductivity [36] or other orders. [28] Second, while the additional electron again creates its own cloud, it also exchanges phonons with the other (bi)polarons, giving rise to effective electron-electron interactions that become considerable at stronger electron-phonon couplings. These strong effective interactions turn the system into a strongly correlated one even if the bare electron-electron interactions are weak.

It is possible to express the sum of the electron-addition and electron-removal propagators as an infinite series of diagrams which in principle are fully known [10] but in practice are difficult to evaluate efficiently if the electron-phonon coupling is strong. Even for weak-to-moderate couplings, an efficient semi-analytical summation is only possible in the Migdal limit, for carrier concentrations $x$ sufficiently large so that the Fermi energy $E_F$ is much larger than the typical phonon energy $\Omega$. Migdal showed that vertex corrections can be ignored when $E_F \gg \Omega$, greatly simplifying the summation of the remaining diagrams. [37] Even this limit, however, cannot be treated accurately for strong electron-phonon couplings $\lambda \sim 1$ where 'polaronic effects' become considerable. Furthermore, it is well established that Migdal's approximation fails when $E_F < \Omega$. [38]

In this Letter we advance the study of electron-addition spectral functions to this un-explored area of the parameter space: low carrier concentrations $x \ll 1$ so that $E_F < \Omega$, at *all couplings*. The results presented here are limited to spinless electrons to avoid the possibility of bipolaron formation and/or superconductivity. [39] As discussed below, we expect these results to also be relevant for spinful electrons [40–45] if the bare electron-electron repulsion is much stronger than the phonon-mediated attraction, preventing bipolaron formation/superconductivity.

Our results are obtained using a generalization of the MA variational approximation to small $x$ concentrations, which describes the formation of a polaron when a fermion is added above an inert Fermi sea. [46] We present MA results in one-dimension (1D) so that we can use density matrix renormalization group (DMRG) –with significant enhancements obtained in Ref. [47] and described in the Methods– as an unbiased computational method to gauge them. This comparison validates the accuracy of the MA fermion addition spectral weights for $x < 0.1$ and $\Omega/t > 0.5$ at all values of $\lambda$. It is important to emphasize that MA is trivial to generalize to dimensions $D > 1$ and its accuracy improves with increasing $D$, [18] therefore it allows the accurate study of polarons in any $D$ for small $x$. The results presented here are for the Holstein model, but DMRG (in 1D) and MA have been used successfully to study other electron-phonon couplings. [20] Thus, our work opens the way to efficiently study polaron properties for a wide variety of models in a new region of the parameter space. Furthermore, valuable lessons are learnt from analyzing the processes that turn out to be most relevant in MA, and may provide the key to expanding polaron studies to even wider regions of the parameter space.

## 2  Model and Results

As advertised, we study the one-dimensional spinless Holstein model:

$$\mathcal{H} = \mathcal{H}_{\mathrm{e}} + \mathcal{H}_{\mathrm{ph}} + V_{\mathrm{e-ph}} \qquad (2)$$

where $\mathcal{H}_{\mathrm{e}} = -t\sum_n (c_n^\dagger c_{n+1} + \mathrm{H.c.})$ describes nearest-neighbor hopping on a chain with lattice constant $a = 1$, and $c_n^\dagger = \sum_k e^{-ikn} c_k^\dagger / \sqrt{N}$ creates an electron at site $n$. Here $N \to \infty$ is the number of sites and $k \in (-\pi, \pi]$ is the crystal momentum. Phonons are described by an Einstein model: $\mathcal{H}_{\mathrm{ph}} = \Omega \sum_n b_n^\dagger b_n$ (we set $\hbar = 1$) where $b_n^\dagger$ creates a a phonon at site $n$. The Holstein electron-phonon coupling is $V_{\mathrm{e-ph}} = g\sum_n c_n^\dagger c_n (b_n^\dagger + b_n)$. In standard fashion, we characterize the effective strength of electron-phonon coupling via the dimensionless coupling $\lambda = g^2/(2\Omega t)$.

The polaron dispersion is defined by the lowest-energy feature in the spectral weight $A(k, \omega) = -\frac{1}{\pi}\mathrm{Im}G(k, \omega + i\eta)$ where $G(k, z)$ is the spinless analog of the propagator in Eq. (1) and $\eta \to 0$ is an artificial broadening. Before discussing MA and the approximations it comprises, it is worth first seeing how it performs. Figures 1 and 2 show the electron-addition spectra when $\Omega = t$ and $0.5t$, respectively, for concentrations $x \in [0, 0.15]$ and couplings $\lambda \leq 1$. We note that our calculations are for a canonical ensemble so the energy $\omega$ is on an 'absolute' scale where $\omega = -2t$ marks the ground-state of a single fermion ($x = 0$). The more customary grandcanonical results are obtained by shifting the curves upwards by the corresponding chemical potential $\mu$, so that fermion addition spectral weight appears only for $\omega \geq 0$. We also note that because MA approximates $|GS_{N_e}\rangle$ with an inert Fermi sea, here one only observes finite electron-addition spectral weight for $|k| \geq k_F$ (the dashed verticals lines mark $k_F$). DMRG confirms that this is accurate for the polaron band, however at higher energies DMRG finds finite (albeit small) fermion addition spectral weight below $k_F$.

The most striking observations are: (i) while a complete polaron band at all $|k| \in [k_F, \pi]$ is observed for the single polaron ($x = 0$) as expected, it also appears for moderate-to-large $\lambda \geq 0.5$ for finite $x$. By contrast, for finite $x$ and small $\lambda = 0.25$, we instead observe the kink at $E_F + \Omega$ predicted by perturbation theory, and also well established in the Migdal limit. The emergence of the complete polaron band at finite $x$ is a signature of the 'polaronic effects', and here it is observed for a much weaker $\lambda = 0.5$ than the expected $\lambda \sim 1$. We further discuss this issue below. (ii) The polaron bandwidth narrowing (signalling a heavier polaron at stronger coupling, all else being equal) decreases with increasing $x$, signalling a lighter polaron at larger $x$ (all else being equal). The significant variation of the effective polaron mass $m^*$ with $x$ even in this narrow range of $x$ values, shows that ignoring this dependence on doping in phenomenological models is very questionable. This is further discussed in Ref. [46]. (iii) For $\lambda = 0.25$, the higher energy spectrum with $\omega > E_F + \Omega$ roughly follows the free particle dispersion with additional broadening due to scattering on free phonons. In contrast, for strong coupling $\lambda = 1$ the higher energy spectrum is split into a series of 'replicas' that are totally unlike the free electron spectrum. This change is also a clear signature of the 'polaronic effects'.

The existence or absence of a polaron state at larger $k$ is rather hard to ascertain from Figs. 1,2 due to the broadening $\eta = 0.05$ used (this value was needed to get a better convergence for the DMRG results, because the Correction Vector method [48–50] used here is known to be numerically ill defined for $\eta \to 0$ [51]). In Fig. 3, we plot the correspondings MA results obtained at $k = 0.8\pi$, $x = 0.1, \Omega = 1$ for several smaller $\eta$; these results are representative for

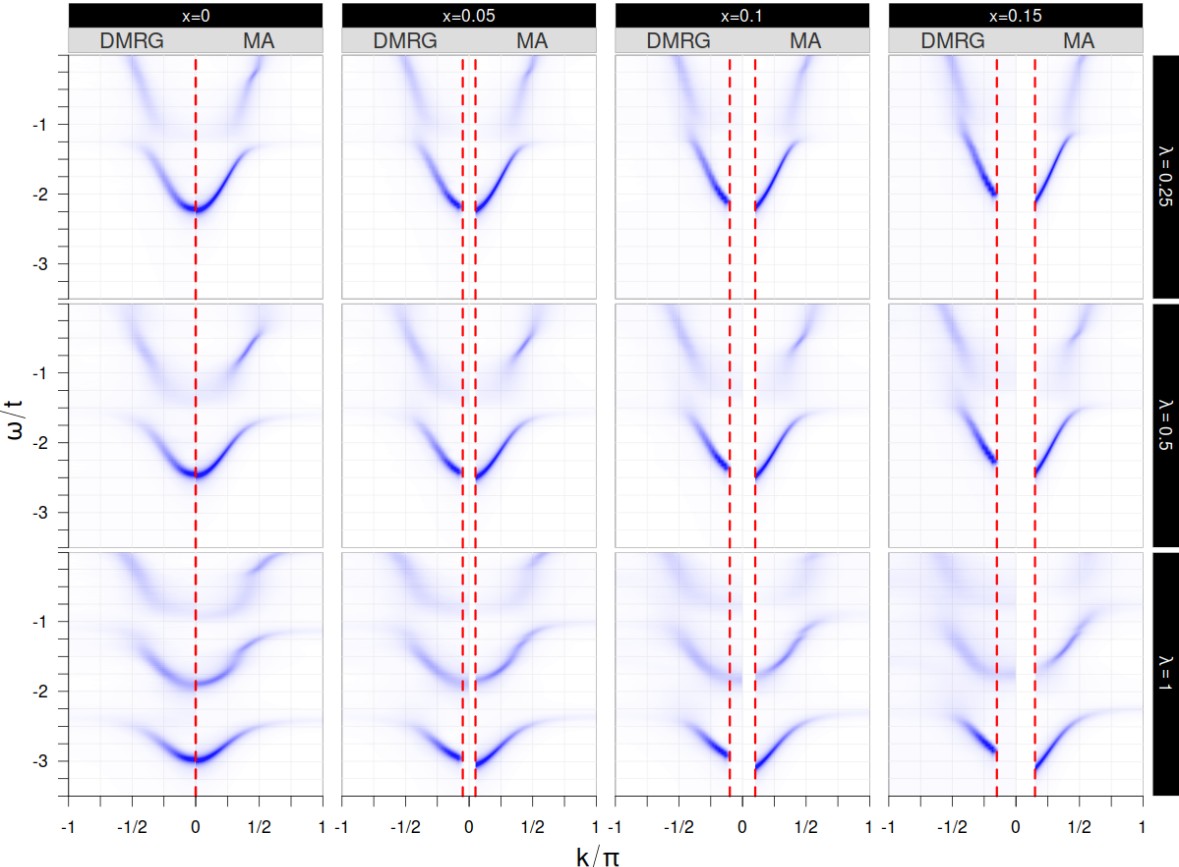

Figure 1: **Electron-addition spectral functions** $A(k, \omega)$ for $\Omega = t$ and $\eta = 0.05$. From left to right, columns show results for $x = 0, 0.05, 0.10, 0.15$, respectively, while rows show $\lambda = 0.25, 0.50, 1.00$ from top to bottom, respectively. Each contour plot shows the DMRG result for $k < 0$ and the MA result for $k > 0$. For $x = 0$ (left column) the single polaron band extends across the entire Brillouin zone for all $\lambda$, moving to lower energies and becoming narrower as $\lambda$ increases. In contrast, for weak coupling $\lambda = 0.25$ and at finite $x$ there is only a kink at $E_F + \Omega$. For moderate and large $\lambda = 0.5, 1$, however, we see an emerging polaron band across at all momenta $|k| > k_F$. The polaron bandwidth renormalization decreases with increasing $x$, pointing to a more mobile quasiparticle. The agreement between MA and DMRG becomes poorer with increasing $x$ but is reasonable for $x \leq 0.1$.

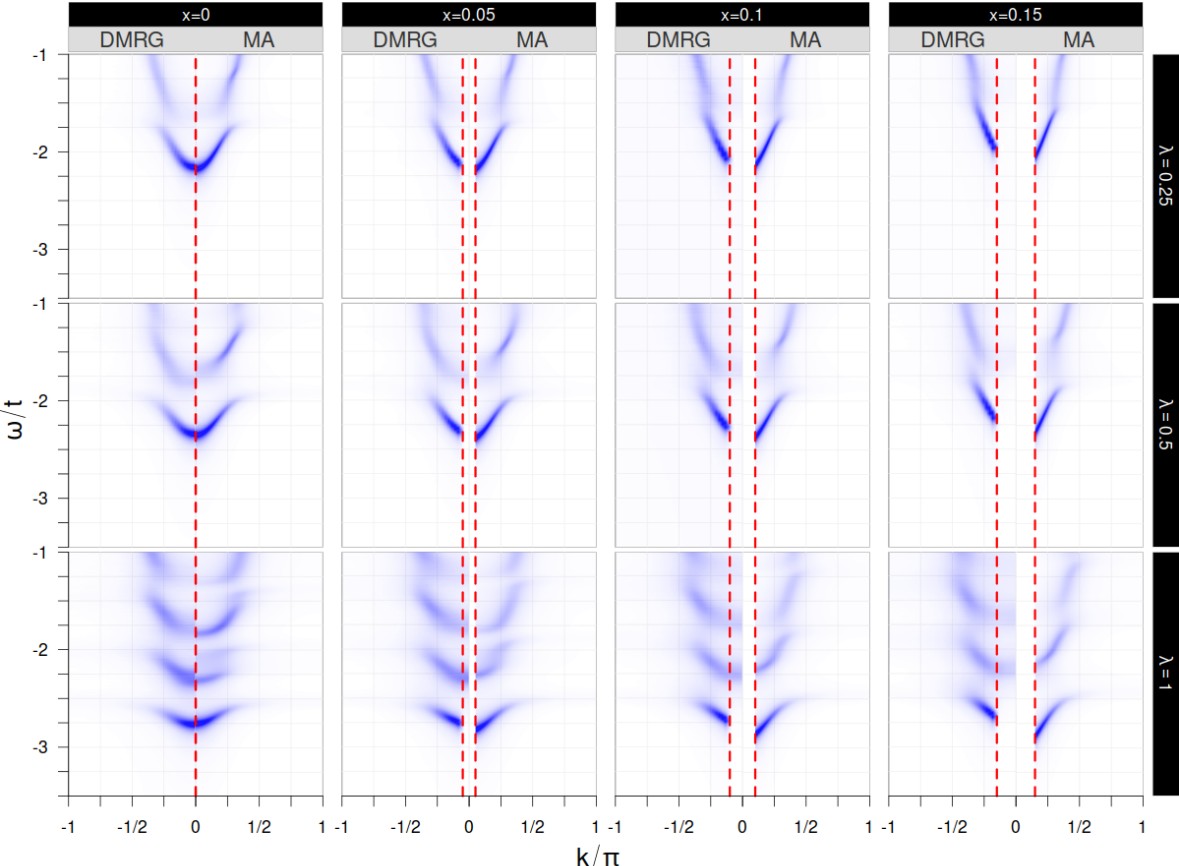

Figure 2: **Electron-addition spectral function** $A(k,\omega)$ for $\Omega = 0.5t$, with everything else as in Fig. 1. The evolution of the polaron band with $x$ and $\lambda$ mirrors the results from Fig. 1, but the quantitative agreement between DMRG and MA is worse than in Fig. 1. For $\lambda = 1$, the polaron band is well separated from the higher energy features and the spectrum is very unlike that of a free electron at all energies. See text for more discussions.

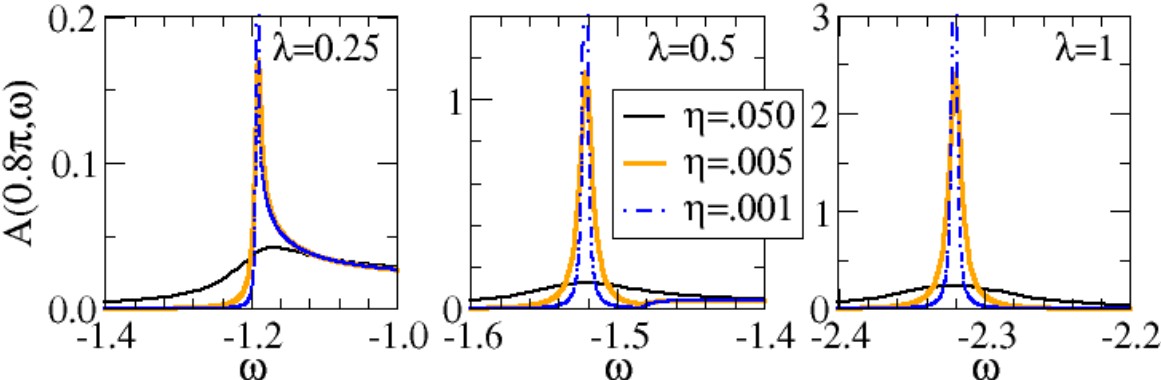

Figure 3: **Emergence of the high-momentum polaron with increasing $\lambda$:** Spectral weight $A(k = 0.8\pi, \omega)$ for $\lambda = 0.25, 0.5$ and 1, respectively, and for different $\eta$ when $x = 0.1, \Omega = 1$. For $\eta = 0.05$ (the value used in Figs. 1,2) the low energy feature looks quite similar in all three cases. The lower $\eta$ results demonstrate that for $\lambda = 0.25$, this feature is the lower edge of a continuum, *i.e.* here there is no polaron state. For $\lambda = 0.5$, this feature resolves into a sharp peak lying below a continuum, demonstrating that a polaron has formed. For $\lambda = 1$ the feature is a single sharp peak, showing the polaron well below higher energy states.

all $k$ near the Brillouin zone edge. They clearly show that while a polaron state is absent in this region for $\lambda = 0.25$, it has already emerged for $\lambda = 0.5$, which is a significantly smaller coupling than expected. Further increasing $\lambda$ pushes the polaron away from the continuum and gives it more quasiparticle weight.

## 3 Discussion

Overall, the agreement between DMRG and MA is qualitatively reasonable at most shown values, becoming quantitatively accurate for $x < 0.1$ and improving for larger $\Omega/t$ ratios; all this is achieved with a trivial MA computational cost (the results for any panel from Figs. 1 or 2 take under one second to generate on a regular workstation). In contrast, for the DMRG calculations, a *single* frequency spectral function calculation can take up to 4 hours to complete on a cluster node (we used an Intel Xeon Gold 6130 CPU node, hyperthreading over 8 cores); trivial parallelization is used to compute multiple frequencies simultaneously.

To understand the lessons that can be learned from these results, we now turn to analyzing the approximations made in MA. The first major approximation in the finite-$x$ MA is to replace the unknown $|GS_{N_e}\rangle$ by the mean-field approximation $|mf\rangle_{N_e} = |FS\rangle \otimes |\tilde{0}\rangle$, where $|FS\rangle = \prod_{|k|<k_F} c_k^\dagger |0\rangle$ is the spinless Fermi sea with $k_F = \pi x$, and $|\tilde{0}\rangle$ is a coherent phonon state describing the uniform lattice distortion induced by the uniform carrier distribution (see Methods for more details). [46] We only allow one phonon cloud to be created when the extra fermion is added, thus our variational method essentially analyzes the existence and the nature of a single polaron above an inert Fermi sea. As such, one can think of it as the analog of Cooper's calculation for the binding of two electrons above an inert Fermi sea, in the presence of a weak attraction. [52] In that problem, all the electrons bind into pairs so one

should use the BCS wavefunction [36] instead of the inert Fermi sea; nevertheless, Cooper's calculation gives the correct binding energy because the BCS state and the Fermi sea only differ within a very narrow strip of width $\Omega$ centered at $E_F \gg \Omega$. Similarly, in our problem all fermions get dressed into polarons and the true ground state is some complicated polaron liquid. Because here $\Omega > E_F$, one would naively expect this polaron liquid to be totally different from an inert Fermi sea, nevertheless we find that the inert Fermi sea approximation works surprisingly well for small $x$.

The reason for this unexpected outcome can be qualitatively understood as follows: consider a polaron with a total momentum $k_T$. Its energy is lowered due to level repulsion between the configuration $c_{k_T}^\dagger |0\rangle$ with energy $\epsilon(k_T) = -2t \cos(k_T)$, and the one-phonon continuum comprising the configurations $c_k^\dagger b_{q=k_T-k}^\dagger |0\rangle$ with energies $\epsilon(k) + \Omega$. For $x = 0$ all the one-phonon configurations are available and the hybridization is strongest with the lowest-energy ones with $k \approx 0, q \approx k_T$. At finite $x$, however, the small $|k|$ fermion states are already occupied due to the formation of other polarons. Hybridization with the one-phonon continuum is therefore partially inhibited, explaining why polarons in the polaron liquid are less dressed than single polarons of the same momentum and hence, why the polaron liquid wavefunction is not as different from the inert Fermi sea as one would naively expect for $E_F < \Omega$. [53] This argument also explains the decreased dressing with increasing $x$ of the polaron with $k > k_F$, [46, 53] shown by Figs. 1 and 2.

The second major approximation in this finite-$x$ MA is to ignore phonon renormalization by not allowing particle-hole pairs to be excited out of the Fermi sea through phonon absorption. Interestingly, this approximation is also used in the Migdal limit, raising the question whether it is justified everywhere in the parameter space. In fact, we can include these processes relatively easily in MA, however for $x < 0.15$ we found that they have hardly any effect on the polaron band (differences are hard to see on the scale of Figs. 1 and 2). Higher energy features are more strongly affected and the agreement with DMRG improves, however not enough, in our opinion, to justify the increased computational cost. We will report these incremental improvements elsewhere.

The third approximation is the same as for the $x = 0$ MA, namely the variational constraints on the phonon configurations included in the polaron cloud. The results in Figs. 1 and 2 are from a one-site MA$^{(2)}$ approximation, [19] which allows the phonon cloud to extend over one site (with arbitrary number of phonons at this site which can be at any distance from the additional fermion); in addition, up to two free phonons not bound to the cloud can be placed anywhere else. We have also implemented a three-site MA$^{(0)}$ approximation, [20] where the phonon cloud is allowed to extend over three consecutive sites and there are no free phonons. The results are qualitatively similar and the quantitative differences for the polaron bands are not significant enough to justify the increased computational cost of the bigger variational space.

Alltogether, this analysis reveals that the main step necessary to improve the agreement for larger $x$ is to find a better description of the $|GS_{N_e}\rangle$ polaron liquid. In other words, we need to find the polaron liquid analog to the BCS wavefunction. Nevertheless, we emphasize that the finite-$x$ MA presented here already works well for small $x$, is easy to implement and is very efficient, and can be straightforwardly generalized to other types of bosons and of couplings as well as to higher $D$, becoming more accurate with increasing $D$. It therefore allows us to explore with confidence a part of the parameter space that was not studied before, and which is relevant for many quantum materials that show interesting behavior at low doping,

whether it is in a nearly empty band (small $x$) or a nearly full band (small $1 - x$).

Finally, we note that *if* the phonon-mediated attraction is small compared to the bare electron-electron attraction so that bipolarons are unstable, this finite-$x$ MA carries essentially without change to spinful fermions. Furthermore, because MA has been generalized to study single bipolarons, [54] we can also study the effect of electron-phonon coupling on two fermions above an inert Fermi sea, generalizing Cooper's calculation away from weak coupling and $\Omega \ll E_F$, and possibly opening the way to understand finite-$x$ bipolaron liquids. This work is under way.

## 3.1   Methods

### 3.1.1   Momentum Average Approximation

The finite-$x$ MA generalization for both electron addition and electron removal propagators is presented in Ref. [46], up to one important correction which we review here. This has to do with the definition of the dressed phonon operators defining the new 'vacuum' $B_i|\tilde{0}\rangle = 0$ (for the original phonons, $|\tilde{0}\rangle$ is a coherent state describing the mean-field uniform lattice distortion induced by the uniform density of fermions). Instead of defining them as $B_i = b_i + gx/\Omega$, [46] one needs to use $B_i = b_i + g\hat{N}_e/(\Omega N)$ where $\hat{N}_e$ is the operator for the total number of fermions. In the ground-state $\hat{N}_e \to N_e = Nx$ and the two definitions agree, however when the extra fermion is added, $\hat{N}_e \to N_e + 1$. This subtle difference is important when considering the contribution from the mean-field shift $-g^2\hat{N}_e^2/(\Omega N)$ to the difference $\mathcal{H} - E_{GS,N_e}$ in the resolvent. In Ref. [46] these energy shifts cancelled out. The proper definition used here leads to a finite difference in the thermodynamic limit $-g^2(N_e+1)^2/(\Omega N)+g^2 N_e^2/(\Omega N) = -2g^2 x/\Omega$ which has a significant effect on the location of the spectral weight for the larger $x$, $\lambda$ values. This correction is trivially accounted for by replacing $z \to z + 2g^2 x/\Omega$ everywhere in the MA expressions provided in Ref. [46].

### 3.1.2   Density Matrix Renormalization Group

We use the DMRG [55] as implemented in the DMRG++ software [56] at zero temperature, to calculate the spectral functions $A(k,\omega)$ using the recently developed DMRG root-N Krylov correction-vector approach [47].

In a nutshell, the root-N Krylov DMRG method evaluates the "Correction-Vector" [48,49] $|CV_{x_c}\rangle = [\omega - \mathcal{H} + E_{GS,N_e} + i\eta]^{-1} c_{x_c}^\dagger |GS_{N_e}\rangle$ by applying $N$ times the root-$N$ progator $[\omega - \mathcal{H} + E_{GS,N_e} + i\eta]^{-1/N}$ to the initial vector $c_{x_c}^\dagger|GS_{N_e}\rangle$, using at each step the standard DMRG Correction Vector Krylov algorithm introduced in Ref. [50]. In this work, we use finite-size chains with open boundary conditions (OBC) and employ the center-site approximation, in which the additional fermion is added at the center site $x_c = N/2$. This approximation reduces the computational cost by order of $N$ and becomes exact in the thermodynamic limit, although it introduces "ringing" artifacts in the spatial Fourier Transform in small finite systems. Indeed, once the Correction Vector is computed, one extracts $O_{i,x_c}(\omega) = -\frac{1}{\pi}\text{Im}[\langle GS_{N_e}|c_i|CV_{x_c}\rangle]$ for all sites $i$. These are then used to calculate $A(k,\omega) = \sum_{i=1}^N \cos[k(r_i - r_c)]O_{i,x_c}(\omega)$. As seen in figs. 1-2, in systems with $N = 80$ lattice sites artifacts from the use of OBC and the center site approximation are minor.

In Ref. [47] it was shown that, at sufficiently large $N$, the DMRG root-$N$ Krylov method improves both the computational speed and frequency resolution (signal-to-noise ratio) of

the spectal functions compared to the standard Correction-Vector DMRG method [48–50]. In particular, the root-N Krylov method is especially useful at large target frequencies with respect to the Fermi energy, where large bond dimensions (or DMRG states) are typically needed. These improvements were essential to obtain the DMRG results shown here. We refer the reader to Ref. [47] for more details.

In this work, we used the standard Fock basis representation of the phonon degrees of freedom, utilizing up to 12 phonon states to represent the local phonon Hilbert space. We therefore did not need to use the more sophisticated local phonon optimization methods [29,57–63] or the recently developed projected purification DMRG method. [64,65] Numerical results were converged with respect to the bond dimension $m$. A maximum $m = 1000$ (and a minimum $m_{\min} = 24$) provides convergence with a truncation error smaller than $10^{-6}$ for the frequency dependent calculations. For the root-$N$ Correction Vector Krylov calculations the choice $N = 20$ has shown the best compromise in terms of moderately large bond dimension required and computational speed. Finally, we set the the Krylov space tridiagonalization error to $\epsilon_{\text{Tridiag}} = 10^{-9}$ in order to avoid the proliferation of Krylov vectors (and thus Lanczos iterations), and their reorthogonalizations. Sec. A provides computational details (input file for the DMRG++ code) to reproduce the DMRG data shown in this work.

## Data availability

The instructions to build the input scripts for the DMRG++ package to reproduce the DMRG results can be found in Sec. A. The DMRG data/scripts to reproduce our figures is available as a public data set at [66]. Raw DMRG data files will be made available upon request.

## Code availability

The numerical results reported in this work were obtained with DMRG++ versions 6.05 and PsimagLite versions 3.04. The DMRG++ computer program [56] is available at https://github.com/g1257/dmrgpp.git, see Sec. A for more details.

## Acknowledgements

We thank A. Feiguin, S. Johnston, N. Prokof'ev and J. Sous for useful comments and suggestions. We acknowledge support from the Max Planck-UBC-UTokyo Center for Quantum Materials and Canada First Research Excellence Fund (CFREF) Quantum Materials and Future Technologies Program of the Stewart Blusson Quantum Matter Institute (SBQMI), and the Natural Sciences and Engineering Research Council of Canada (NSERC). This work used computational resources and services provided by the Advanced Research Computing at the University of British Columbia. MB acknowledges the hospitality of the Leibniz Institute for Solid State and Materials Research (IFW) Dresden, where part of this work was carried out.

# Author contributions

M. Berciu conceived the project and performed the MA calculations. A. Nocera performed the DMRG calculations. M. Berciu and A. Nocera wrote the manuscript.

# Competing interests

The authors declare no competing interests.

Correspondence and requests for materials should be addressed to A. Nocera and/or M. Berciu.

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

# A  Computational details to reproduce the DMRG results

Here we provide instructions on how to reproduce the DMRG results used in the main text. The results reported in this work were obtained with DMRG++ versions 6.05 and PsimagLite versions 3.04. The DMRG++ computer program [56] can be obtained with:

```
git clone https://github.com/g1257/dmrgpp.git
git clone https://github.com/g1257/PsimagLite.git
```

The main dependencies of the code are BOOST and HDF5 libraries. To compile the program:

```
cd PsimagLite/lib; perl configure.pl; make
cd ../../dmrgpp/src; perl configure.pl; make
```

The DMRG++ documentation can be found at  `https://g1257.github.io/dmrgPlusPlus/manual.html` or can be obtained by doing `cd dmrgpp/doc; make manual.pdf`. In the description of the DMRG++ inputs below, we follow very closely the description in the supplemental material of Ref. [47], where similar calculations were performed.

The spectral function results for the 1D Holstein model for $L = 80$ sites, $N = 8$ electrons (thus $x = 0.1$), $\Omega = t$ and $\lambda = 0.25$ can be reproduced as follows. We first run `./dmrg -f inputGS.ain -p 12` to obtain the ground state wave-function and ground state energy with 12 digit precision using the `-p 12` option. The `inputGS.ain` has the form (this is provided at [66])

```
##Ainur1.0
TotalNumberOfSites=160;
NumberOfTerms=3;

### \sum_{<i,j>} -t * (c^\dag_i c_j + h.c.)
gt0:DegreesOfFreedom=1;
gt0:GeometryKind="ladder";
gt0:GeometryOptions="ConstantValues";
gt0:dir0:Connectors=[1.0];
gt0:dir1:Connectors=[0.0];
gt0:LadderLeg=2;

### bosonic hopping \sum_{<i,j>} -t_B * (b^\dag_i b_j + h.c.)
gt1:DegreesOfFreedom=1;
gt1:GeometryKind="ladder";
gt1:GeometryOptions="ConstantValues";
gt1:dir0:Connectors=[0.0];
gt1:dir1:Connectors=[0.0];
gt1:LadderLeg=2;

### electron-phonon interaction \sum_i g*c^\dag_i c_i * (b^\dag_i+b_i) g=sqrt(2*Omega*lambda
gt2:DegreesOfFreedom=1;
gt2:GeometryKind="ladder";
gt2:GeometryOptions="ConstantValues";
gt2:dir0:Connectors=[0.0];
```

```
gt2:dir1:Connectors=[0.707];
gt2:LadderLeg=2;

Model="HolsteinSpinlessThin";
SolverOptions="twositedmrg,CorrectionTargeting,vectorwithoffsets,useComplex";
InfiniteLoopKeptStates=24;
FiniteLoops=[[79, 1000, 0],
[-158, 1000, 0],
[158, 1000, 0]];
# Keep a maximum of 1000 states, but allow SVD truncation with
# tolerance 1e-12 and minimum states equal to 24
TruncationTolerance="1e-12,24";
# Symmetry sector for ground state N_e = 8
TargetElectronsTotal=8;
# Associated with CorrectionTargeting: noise strength added to
# reduced density matrix to avoid the algorith gets stuck;
CorrectionA=0.01;
OutputFile="dataGS_L80_N8_nph8";
```

The next step is to calculate dynamics for the $A(\mathbf{k}, \omega)$ spectral function using the saved ground state as an input. It is convenient to do the dynamics run in a subdirectory `Aqw`, so create this directory first, and then add/modify the following lines in `inputAqw.ado` (this input is provided at [66])

```
# The finite loops now start from the final loop of the gs calculation.
# Total number of finite loops equal to N+2, here N=6
FiniteLoops=[
[-158, 1000, 2],[158, 1000, 2],
[-158, 1000, 2],[158, 1000, 2],
[-158, 1000, 2],[158, 1000, 2],
[-158, 1000, 2],[158, 1000, 2]];

# Keep a maximum of 1000 states, but allow SVD truncation with
# tolerance 1e-6 and minimum states equal to 24
TruncationTolerance="1e-6,24";

# The exponent in the root-N CV method
CVnForFraction=6;

# Tolerance for Tridiagonal Decomposition of the effective Hamiltonian
TridiagonalEps=1e-12;

# Solver options should appear on one line, here we have two lines because of formatting pur
SolverOptions="useComplex,twositedmrg,vectorwithoffsets,
               TargetingCVEvolution,restart,fixLegacyBugs,minimizeDisk";

# RestartFilename is the name of the GS .hd5 file (extension is not needed)
RestartFilename="../dataGS_L80_N8_nph8";
```

```
# The weight of the g.s. in the density matrix
GsWeight=0.1;
# Legacy, set to 0
CorrectionA=0;
# Fermion spectra has sign changes in denominator.
# For boson operators (as in here) set it to 0
DynamicDmrgType=0;
# The site(s) where to apply the operator below. Here it is the center site.
TSPSites=[79];
# The delay in loop units before applying the operator. Set to 0
TSPLoops=[0];
# If more than one operator is to be applied, how they should be combined.
# Irrelevant if only one operator is applied, as is the case here.
TSPProductOrSum="sum";
# Sets the number of sweeps to 1 before advancing in "time"=1/N
TSPAdvanceEach=78;
# How the operator to be applied will be specified
string TSPOp0:TSPOperator=expression;
# The operator expression to apply the c^\dagger operator on the center site
string TSPOp0:OperatorExpression="c?0'";
# How is the freq. given in the denominator (Matsubara is the other option)
CorrectionVectorFreqType="Real";
# This is a dollarized input, so the
# omega will change from input to input.
CorrectionVectorOmega=$omega;
# The broadening for the spectrum in omega + i*eta
CorrectionVectorEta=0.05;
# The algorithm
CorrectionVectorAlgorithm="Krylov";
#The labels below are ONLY read by manyOmegas.pl script
# How many inputs files to create
#OmegaTotal=40
# Which one is the first omega value
#OmegaBegin=-4.0
# Which is the "step" in omega
#OmegaStep=0.1
# Because the script will also be creating the batches,
# indicate what to measure in the batches
#Observable=c?0
```

Notice that the main change with respect of a standard CV method input is given by the option `TargetingCVEvolution` in the SolverOptions instead of `CorrectionVectorTargeting`, and the addition of the line `CVnForFraction=6;` We note also that the number of finite loops must be at least equal to the number equal to the exponent in the root-$N$ CV method. As in the standard CV approach, all individual inputs (one per $\omega$ in the correction vector approach) can be generated and submitted using the `manyOmegas.pl` script which can be found in the

`dmrgpp/src/script` folder (but also provided at [66]):

```
perl manyOmegas.pl inputAqw.ado batchTemplate.pbs <test/submit>.
```

It is recommended to run with `test` first to verify correctness, before running with `submit`. Depending on the machine and scheduler, the `BatchTemplate` can be e.g. a PBS or SLURM script. The key is that it contains a line `./dmrg -f $$input "<gs|$$obs|P1>" -p 12` which allows `manyOmegas.pl` to fill in the appropriate input for each generated job batch. After all outputs have been generated,

```
perl myprocAkw.pl inputAqw.ado
```

can be used to process (this script is also provided at [66]) and generate a data file `outSpectrum.c?0.gnuplot` ready to be plotted (using the Gnuplot software, for example).