# Peer review of "Electron addition spectral functions of low-density polaron liquids"

_SciPost Physics_

## Round 2 · Referee Report · Anonymous (Referee 1) · 2023-5-31

Strengths

Clear and concise demonstration and explanation of the performance of the new "Momentum Average" method

Weaknesses

Few new insights about polaron physics

Report

The authors calculate the spectral functions of the one-dimensional spinless Holstein model at low band filling using two methods, a new kind of
variational method called Momentum Average (MA) proposed recently by one of them (ref. 46 in the manuscript) and the density matrix renormalization group (DMRG) method. They show that the results agree qualitatively for various parameter regimes and even quantitatively in the low-density weak-coupling limit. This demonstrates the accuracy and potential of the new and simpler MA method in that regime.

The aim of this paper is to demonstrate and explain the surprisingly good
performance of the MA method. The numerical results are plausible and support the conclusions. The discussion of the MA method performance in the third section is very instructive. Overall the paper is very well written, clear and concise. Thus I think that the acceptance criteria are met.

However, the paper provides very little new knowledge about polaron physics. Although figures 1 and 2 compare a lot of data in a very compact way, I am somewhat disappointed after reading the manuscript by the limited amount of information shown. Adding more results would not only help readers to better understand the agreement (and differences) between both methods but could also increase the manuscript significance for polaron physics. For instance, one could compare some MA and DMRG line shapes directly in a figure like fig. 3. Also it would be interesting to compare and discuss derived quantities such as density of states and momentum distributions obtained with MA and DMRG.
  • validity: high
  • significance: good
  • originality: good
  • clarity: top
  • formatting: excellent
  • grammar: excellent

Author:  Alberto Nocera  on 2023-07-18  [id 3818]

(in reply to Report 1 on 2023-05-31)
Category:
answer to question
reply to objection

We thank the Referee for their time and effort reviewing our manuscript, for acknowledging the high-quality of our work and recommending its publication.

The Referee suggested comparing directly the lineshapes predicted by DMRG vs MA. This is done in the next two Figures, where we show low momentum cuts (k = 0.1π − 0.5π) of the spectral functions for MA (solid black line) and DMRG (dashed red lines) for various x, λ. First, we mention that the apparent disagreement for x = 0.10 for the cut at k = 0.1π = kF is because MA has a discontinuity at kF , predicting zero spectral weight below kF and finite spectral weight above kF . The agreement for this specific data set would be much better if we ’integrated’ the MA result for a small range of momenta. Otherwise, we note that DMRG spectral peaks are slightly wider than those obtained in the MA approximation (we highlight that the same η = 0.05t was used in both methods). For finite electronic concentrations, we ascribe this discrepancy to the fact that spectral weight in DMRG is allowed for |k| < kF while this is not the case in the MA approach. Interestingly, for Ω = 0.5t, Fig. 2 shows less discrepancies in the two methods.

If the Referee(s) recommend it, we can certainly add these plots in the manuscript, although they do not show any new data that is not already there. We agree with the Referee that it would be more instructive if DMRG data could be added in Fig. 3, however it is currently essentially impossible to converge DMRG for the smaller η values. As suggested by the Referee, we computed with DMRG the ground state momentum distributions fuctions for electron addition N + (k) = ⟨ck c†k ⟩, and electron removal N − (k) =⟨c†k ck ⟩ = 1 − N + (k). We included the latter in the manuscript (new Fig. 4) and commented on its differences compared to the step function describing the ’inert Fermi sea’ assumed by MA. This new plot is certainly a clear illustration of shortcomings in the assumed GS used in MA, and might give us, or others, ideas on how to further improve upon it.

Finally, we would like to reply to the statement that our work ’provides little new knowledge about polaron physics’. Respectfully, we disagree. The results for x ≪ 1 have to evolve continuously from the well-known x = 0 single-polaron results, so one cannot expect anything too surprising in this limit. The point of our work is to demonstrate that we can now quantify precisely what these differences are, whether they happen to be small or more substantial; in particular, we show that the rather simple MA generalization performs very well at this task. Such a quantitative description was not available prior to this work. We also think that the fact that the full polaron band emerges for λ < 0.5 at finite x is very surprising (and the second Referee seems to agree). In the Migdal limit this is assumed to happen for λ ∼ 1 however the actual limit of validity is not well understood, and our work suggests that that bound might be rather suspect. We added a short sentence in the manuscript to further emphasize this fact.

Summary of Changes 1. We added Figure 4 showing momentum distribution functions calculated with DMRG. 2. We provided line cuts comparisons between MA and DMRG. In our opinion these do not add new information, however we are happy to include them in the manuscript if the Referee(s) request it.

Attachment:

referee_response1.pdf

---

## Round 2 · Referee Report · Anonymous (Referee 2) · 2023-6-1

Strengths

1) the appearance of the ‘polaronic effects’ persists at finite x already in the intermediate coupling regime; 2) the polaron effective mass decreases with increasing x.  3) in the discussion, authors provide an intuitive mechanism for the latter effect in terms of the filled ‘Fermi sea’ that inhibits hybridization of the injected fermion at finite kT with the states comprised of a product of a free electron at k and the one-phonon at q = kT - k in case k<kF since latter states form the polaron liquid.

Weaknesses

The method has some limitations: 1) it is valid in the low-doping regime, 2) the method can not treat (with some exceptions) attractive interaction between electrons

Report

Considering brief descriptions of both methods where authors present some obstacles they had to overcome during the development of the method, this manuscript is well written. It presents a new method to treat polaronic systems at low doping. In contrast to DMRG, where generalization to higher dimensions is highly non-trivial, this is not the case with the MA method. I strongly support the publication of this manuscript.

Requested changes

If using MA it is possible to extract the effective mass at finite x more quantitatively, I suggest, that authors present a plot depicting the effective mass vs. doping for two (or more) Einstein frequencies and coupling strengths.

  • validity: top
  • significance: high
  • originality: top
  • clarity: top
  • formatting: excellent
  • grammar: excellent

Author:  Alberto Nocera  on 2023-07-18  [id 3817]

(in reply to Report 2 on 2023-06-01)
Category:
answer to question
pointer to related literature

We thank the Referee for their time and effort reviewing our manuscript, for acknowledging the high-quality of our work and for recommending its publication.

We thank the Referee for this suggestion. We note that this analysis was already performed by one of us in Figures 5 and 6 of J. Phys. Mater. 5, 044002 (2022). We included these figures here for the Referee's convenience. Given that this data was already published in an open access journal, we believe that repeating it here would be inappropriate. We have added text in the manuscript to make it more clear to the readers that these results for the effective mass are available elsewhere.

Attachment:

referee_response2.pdf

---

## Editorial Decision

resubmitted